# Prioritizing supports and services to help older adults age in place: A Delphi study comparing the perspectives of family/friend care partners and healthcare stakeholders

Megan Campbell[1], Tara Stewart[2], Thekla Brunkert[3], Heather Campbell-Enns[4], Andrea Gruneir[5], Gayle Halas[1], Matthias Hoben[6], Erin Scott[1], Adrian Wagg[7], Malcolm Doupe[1]*

1 Max Rady College of Medicine, Rady Faculty of Health Sciences, University of Manitoba, Winnipeg, MB, Canada, 2 Department of Community Health Sciences, George and Fay Yee Centre for Healthcare Innovation, University of Manitoba, Winnipeg, MB, Canada, 3 University Department of Geriatric Medicine FELIX PLATTER, Basel, Switzerland Institute of Nursing Science, Department Public Health, University of Basel, Basel, Switzerland, 4 Department of Psychology, Canadian Mennonite University, Winnipeg, MB, Canada, 5 Faculty of Medicine and Dentistry, University of Alberta, Edmonton, AB, Canada, 6 Faculty of Nursing, University of Alberta, Edmonton, AB, Canada, 7 Department of Medicine, Faculty of Medicine and Dentistry, University of Alberta, Edmonton, AB, Canada

* MalcolmBray.Doupe@umanitoba.ca

## Abstract

### Background

Aging in place (AIP) is a policy strategy designed to help older adults remain in their community. While planners internationally have modified aspects of the older adult care continuum (e.g., home care, assisted living, nursing homes) to facilitate AIP, further improvements to community-based supports and services are also required. This study compared and constrasted the community-based factors (e.g., supports, services and personal strategies or characteristics) that family/friend care partners and healthcare stakeholders (i.e., planners/providers) view as most important to help older adults successfully AIP.

### Methods

An initial list of factors shown to influence AIP was created from the academic literature. These factors were used to develop a Delphi survey implemented separately on care partners and healthcare stakeholders. Respondents rated the importance of each factor using a 10-point Likert Scale (1 = not important; 10 = absolutely critical). Consensus in each group was defined when at least 80% of participants scored a factor ≥8 ("very important"), with an interquartile range ≤2. Respondents suggested additional factors during Delphi round one.

### Results

Care partners (N = 25) and healthcare stakeholders (N = 36) completed two and three Delphi rounds, respectively. These groups independently agreed that the following 3 (out of 27) factors were very important to help older adults age in place: keeping one's home safe,

**Data Availability Statement:** All relevant data are within the manuscript and its Supporting Information files.

**Funding:** Funding to conduct this research was provided the Canadian Institutes of Health Research (https://cihr-irsc.gc.ca/e/193.html), application number 402869; MD as Nominated PI. Partial matched funding was provided by Research Manitoba (https://researchmanitoba.ca) and Alberta Health (https://www.alberta.ca/health. aspx). The funders had no role in study design, data collection and analysis, decision to publish, or preparation of the manuscript.

**Competing interests:** I have read the journal's policy and the authors of this manuscript have the following competing interests: Andrea Gruneir is currently an Academic Editor with PLOS ONE.

maintaining strong inter-personal relationships, and coordinating care across formal providers. While healthcare stakeholders did not reach consensus on other factors, care partners agreed that 7 additional factors (e.g., access to affordable housing, having mental health programs) were important for AIP.

## Conclusions

Compared to healthcare stakeholders, care partners felt that more and diverse community-based factors are important to support older adults to successfully AIP. Future research should replicate these findings in other jurisdictions, examine the availability and accessibility of the priority factors, and develop sustainable solutions to enhance their effectiveness.

## Background

Healthcare planners internationally have been searching for the right mix of aging in place (AIP) practices designed to delay or prevent nursing home admission, while supporting older adults to remain in their community for as long and safely as possible [1–4]. Planners across Europe [5], the United States [6], and Canada [7, 8], have used various strategies to de-emphasize institutional care while extending the type and number of home and community-based supports and services that they provide. Further adapting and refining these AIP strategies is important given population aging coupled with the projected costs of providing continuing care services (e.g., home care, assisted living, nursing homes) [6, 9, 10], and people's desire to remain in their own homes or community for as long as realistically possible [11–13].

AIP strategies may focus on restructuring the continuing care system (e.g., increasing publicly funded home care services, creating assisted living to supplement nursing home beds, providing respite to support family and friend care partners) [3, 5, 14], and/or seek to improve and extend allied health and community-based supports. Examples of the latter include comprehensive case management services (e.g., to care for people with dementia), restorative efforts designed to improve one's physical function, and caregiver education and training programs [15–17]. While researchers have shown the benefits of community-based AIP interventions in randomized controlled trials [15, 18], their effect in real-world settings is less clear [19]. Additionally, despite the potential for AIP to promote feelings of dignity and independence [20, 21], several authors have shown that community-living older adults still experience significant challenges related to housing, finances, and safety [22]; maintaining social connections [23]; and accessing healthcare services [24].

As planners strive to improve AIP, further information is needed to help prioritize the kinds of community-based supports and services that older adults require. Family/friend care partners contribute substantially to older adult care [17, 24], and given this expertise, can help to meaningfully guide the design and delivery of services. Research shows that family/friend care partners and healthcare planners/providers often have different priorities regarding older adult care [25, 26], and that innovations reflecting these different perspectives can lead to improved health outcomes and a greater real-world impact [27, 28]. The purpose of this study was to compare and contrast the community-based factors (e.g., supports, services and personal strategies or characteristics) that family/friend care partners and healthcare stakeholders (i.e., planners and providers) view as most important to help older adults successfully AIP, safely and with as high a quality of life as possible. Study results can help to further establish community-based AIP priorities and provide future research directions.

## Methods

### Overview and definitions

This study was conducted using a modified Delphi method. This approach generates consensus-based expert opinion using an iterative questionnaire and feedback process [29]. The Delphi method has previously been used to develop guidelines that help families care for people with dementia [30], to assess reportable and preventable events in home care [31], to prioritize quality measures of nursing home care [32–34], and to identify operational standards and guidelines for palliative care [35]. Similar to the approach used by others [30, 36], we conducted the Delphi survey independently with two expert groups (care partners and healthcare stakeholders), enabling us to compare and contrast the priorities identified by these different expert groups. The following definitions were used to guide this research.

**Aging in place.**   Guided by the Canadian federal and provincial governments responsible for seniors [3], we defined aging in place as any strategy designed to help people live safely and independently in their home or community for as long as possible. The term 'community' in this research includes home living with or without home care services, and congregate housing options such as assisted living [3, 37], but excludes people residing in nursing homes and chronic care hospitals.

**AIP factors.**   Based on our analyses of the existing literature [38–45] and input from our team of community advisors and healthcare planners, we defined these factors as any community-based supports, services, personal strategies or characteristics that enable older adults to successfully age in place. Examples include (a) policies to ensure affordable housing, (b) medical services that provide timely and necessary care, and (c) personal approaches used to keep physically, mentally, and/or socially well. For the purposes of this research, we excluded from this definition the health and social services provided to assisted living residents (e.g., staffing composition and levels, admission guidelines, types of care provided).

**Family/Friend care partners.**   This term refers to people who provide non-professional care or support to an older adult friend or family member.

### Study setting

This study is part of a larger research program funded by the Canadian Institutes of Health Research [46] designed to compare continuing care health service policies, practices and utilization patterns between Edmonton, Alberta and Winnipeg, Manitoba. While Canada has a national insurance plan that funds medically necessary physician and hospital services in each province [47], planners have more latitude when determining the type and volume of continuing care services that are delivered provincially. Manitoba and Alberta were selected for the broader research program given their similar underlying population characteristics (e.g., household size, age structure, prevalence of caregiving) [48, 49], their long-term commitment to support AIP [50, 51], coupled with their differing continuing care health services (e.g., compared to Alberta, Manitoba has 21% more nursing home beds per capita 85+ year old [52], and one versus three levels of community-based assisted living care [37, 53]). Delphi respondents in the present study were recruited from these regions.

### Framework

This research was guided by the Lau et al. (2007) Health-Related Safety Framework which shows how micro, mezzo, and macro-level factors can minimize preventable and unintended harm to community-dwelling older adults resulting from breakdowns in the societal system [54]. Micro factors include an individuals's biological or psychological characteristics (e.g., physical, mental and functional health; attitude and knowledge; personal health behaviours).

Mezzo factors include one's social network, the structure and features of their home and/or community, and available medical and social services. Macro factors include the broader economic, societal and political forces that may affect a person's ability to age in place (e.g., having access to funds that help people purchase assistive technology) [54]. Similar to others [55, 56], we used the Health-Related Safety Framework to (1) ensure that Delphi responses were based on a diverse array of community-based AIP factors, (2) help structure our study findings.

## Selecting factors to include in the Delphi survey

This research was conducted by a team of researchers, healthcare planners, and community advisors (i.e., care partners with system experience) from each province. Two members of our research team (MC and ES) used key words from select articles and reports [38–42] to develop and apply the following search terms to select databases (Google Scholar, PubMed, CINAHL).

> *"older adults"* AND *"aging in place"* AND/OR *"community supports"; "older adults"* AND *"facilitators for aging in place"; "delay"* OR *"prevent"* AND *"nursing home admission"*

Titles and abstracts were reviewed, and potential factors were extracted from relevant articles. MC, ES, HCE and MD amalgamated these factors into a comprehensive list and provided a lay description of each factor for use in the Delphi survey. As part of a full-day workshop, our entire team reviewed and refined the list of factors and their definitions.

## Participant recruitment, survey development and application

This research was approved by the University of Manitoba Health Research Ethics Board (reference number HS22703 (H2019:117)), the Winnipeg Regional Health Authority Research Access & Approval Committee (#2018–027), the University of Alberta Health Research Ethics Board (MS3_Pro00089982), and Alberta Health Services. Informed consent was obtained from all participants through the online survey.

Two groups of study participants – family/friend care partners ('care partners') and healthcare stakeholders (i.e., planners and providers) – were recruited in June and July of 2019 with assistance from team members. To be included, care partners must have lived in Manitoba or Alberta, and have had current or previous experience providing care. These participants were recruited through personal contacts made by our community advisors, and also through Manitoba's and Alberta's Primary and Integrated Healthcare Innovation Networks, which help researchers to develop patient partnerships [57, 58]. Community advisor team members approached their personal networks and identified individuals who met the study criteria. MC followed up with interested individuals to confirm their eligibility, answer questions, and identify additional participants using snowball sampling.

To be included in the study, healthcare stakeholders must have had experience working in the Winnipeg or Edmonton continuum of healthcare services, with an emphasis on older adult community-based care. Those who worked exclusively in nursing homes were excluded. Healthcare planner team members identified colleagues using these criteria; MC followed up with each potential participant to determine their interest, confirm eligibility, answer questions, and to identify additional participants using snowball sampling.

A draft version of the survey was piloted with a total of four team members (two healthcare planners, two community advisors). These team members completed the survey and provided feedback about its format and content. They also completed a revised version of the survey to verify that the appropriate improvements were made. The final version of the Delphi survey was administered online using SimpleSurvey [59], separately to each participant group (S1 File

contains the round 1 survey for healthcare stakeholders). The survey first explains the study goals, gives key definitions, and provides directions for completing the questionnaire, including a scoring example. Participants were then instructed to complete a consent section, a participant information section (documenting their type and duration of experiences), and the actual questionnaire. With the exception of the group-specific participant information questions, both participant groups completed the same survey.

At the end of Round 1, participants were asked to suggest additional factors to include in subsequent survey rounds. To facilitate group comparisons, the suggestions made by each group were provided to the other. In Round 2, participants from each group were provided with their Round 1 score for each factor, the median score provided by peers, and an anonymized list of (group-specific) comments used to rationalize people's scoring choice. Participants were asked to re-score each factor taking this information into account. Those who did not complete round one were ineligible to participate in subsequent Delphi rounds.

## Delphi scoring and data analysis

Care partners and healthcare stakeholder participants completed the Delphi survey independently, and hence scoring and analyses were conducted separately for each group. Participants in each group rated every factor from 1 ("not important; by providing this score, you are saying that a factor does not help older adults to live successfully the community") to 10 ("absolutely critical; this factor is amongst the top 1 or 2 things you feel are needed to help most older adults live successfully in the community"). A score of '8' defined factors that were considered 'very important' ("this factor is amongst the top 5 or 6 things you feel are needed to help most older adults live successfully in the community").

Results from each Delphi round were analyzed as per the method described by van der Steen et al. (2014), separately for care partners and healthcare stakeholders [60]. Measures of central tendency (median) and variation (interquartile range, IQR) were used to define the following levels of agreement:

*Very High* (80[+]% of participants provided a score ≥8; IQR = 0);

*High* (80[+]% of participants provided a score ≥8; IQR = ≤2);

*Moderate* (60[+]% of participants provided a score ≥8; IQR ≤4); and

*Low* (all other results).

Consensus on a given Delphi factor was defined when participants reached a 'high' or 'very high' level of agreement. As per Jorm (2015), once consensus was reached for a factor, we removed it from subsequent survey rounds [61]. Care partners reached consensus on 10 factors after two Delphi rounds, at which point we decided not to conduct a third Delphi round for this group. This decision is consistent with other studies [36], supports our study goal to identify only the most important AIP factors, and aligns with our rating instructions (i.e., to give, at most, 5 or 6 factors a score of '8'). Group specific results were analyzed separately, enabling us to compare and contrast the priotities identified by each expert group.

## Results

### Community-based aging in place factors

Twenty-three factors were extracted from the academic literature, and 4 additional factors were added by participants during the first Delphi round, for a total of 27 factors examined in

this research (Table 1). These factors were categorized into the micro/person (n = 8), mezzo/community (n = 15), and macro/policy and societal (n = 4) levels.

## Study participants

Twenty-five care partners and 36 healthcare stakeholders participated in this study with low dropout between rounds (Table 2). Care partners were predominantly female (96%) and were on average 67 years old. Forty percent of these participants were currently providing care at the time of the study, and 60% had received either formal or informal care at some point in their life.

**Table 1. Survey factors by source, and categorized by the Health-Related Safety Framework [54].**

| MICRO | | | MEZZO | | | | MACRO |
|---|---|---|---|---|---|---|---|
| Biological | Psychological | Other | Social Network | Home and Neighbourhood Structure | Social Services | Medical Services | Policy and Societal |
| Keeping physically active—Grimmer et al. 2015 [43] | Not having significant behavioural or mental health disorders -Luppa et al. 2009 [42] | Being someone who prepares and plans for the future (e.g., participates in health promotion activities, plans financially for the future, develops new skills)—Scharlach et al 2016 [39] | Having strong relationships and links with family, friends, and the community—Grimmer et al. 2015 [43] | Living in a safe home environment (e.g., with enough safety aids and equipment)—Grimmer et al. 2015 [43] | Having accessible and affordable community-based services (e.g., adult education, recreation and support programs)—Cao et al. 2016 [44] | Having coordinated care between all types of formal health care providers (for example, physicians, home care workers, social workers)—Brown et al. 1997 [45] | Having access to affordable housing—Summer, 2005 [40] |
| Being continent (with or without the use of continence aids)—Friedman et al. 2005 [41] | Thinking of oneself as healthy—Luppa et al. 2009 [42] | Having enough money to afford to stay successfully in the community—Scharlach et al 2016 [39] | | Having a home layout that is appropriate (e.g., the absence of stairs)—Scharlach et al 2016 [39] | Having a resource (e.g., information call centre) that helps people make informed choices about health care services that are available to them—Summer, 2005; Grimmer et al. 2015 [40, 43] | Having physicians who provide house-calls & home visits—Brown et al. 1997 [45] | Having policies that allow people to reside in the community with an acceptable level of risk—Fancey & Keefe, 2014 [38] |
| | Maintaining a positive attitude, having a high self-esteem and/or sense of personal identity—Grimmer et al. 2015 [43] | | | | Having public transportation that is affordable, reliable and accessible—Grimmer et al. 2015 [43] | Having medical professionals (e.g., nurse practitioners, pharmacists) who regularly check the # and type of medications people are taking—Luppa et al. 2009 [42] | Having access to funds that help people purchase assistive technology (e.g., motorized wheelchairs) and/or to modify their home (e.g., put in a wheelchair ramp)—Scharlach et al 2016; Summer, 2005 [39, 40] |
| | Keeping mentally active—Grimmer et al. 2015 [43] | | | | Having formal healthcare providers (e.g., physicians, home care workers) who are aware of community-based services—Fancey & Keefe, 2014 [38] | Having good communication between informal & formal caregivers—Brown et al. 1997 [45] | Ensuring that community-based alternatives to nursing home use (e.g., supportive housing in Manitoba, lodge and supportive living in Alberta) are affordable* |
| | | | | | Having training & education programs for informal caregivers—Friedman et al. 2005 [41] | Ensuring that people have adequate access to important allied and medical services (e.g., glasses, dental care, affordable medications, physiotherapy)* | |
| | | | | | Having programs that help people to cope with mental health challenges (e.g., anxiety, depression, loneliness)* | | |
| | | | | | Having programs that provide support to complete household chores (e.g., shoveling, mowing grass, completing minor household repairs) and other daily tasks (e.g., banking, grocery shopping)* | | |

* Factor was added after Delphi Round 1.

**Table 2. Description of Delphi participant characteristics.**

| Family/Friend Care Partners (N = 25)[†] | |
|---|---|
| | **n (%)** |
| **Profile** | |
| Participant Sex (Female) | 24 (96%) |
| Age in Years, Average (SD) | 67 (11.9) |
| Number of people ≥ 65 years old | 17 (68%) |
| Region (Winnipeg, Manitoba) | 13 (52%) |
| **Care Partner Status**[*] | |
| Past Care Partner | 17 (68%) |
| Present Care Partner | 10 (40%) |
| **Care Receiving Status**[*] | |
| Previously received formal care | 6 (24%) |
| Previously received unpaid care from family or friends | 9 (36%) |
| **Response Rate (based on 28 recruited participants)** | |
| Round 1 | 25 (89%) |
| Round 2 | 24 (85.7%) |
| **Healthcare Stakeholders (N = 36)[‡]** | |
| | **n (%)** |
| **Profile** | |
| Participant Sex (Female) | 30 (83.3%) |
| Age in Years, Average (SD) | 45.6 (10.4) |
| Region (Winnipeg, Manitoba) | 24 (66.7%) |
| **Years of related working experience, Average (SD)** | 15.8 (9.4) |
| **Job title** | |
| Facility Level Planner or Provider | 8 (22.2%) |
| Regional Planner | 18 (50.0%) |
| Government Policy Maker | 10 (27.8%) |
| **Education** | |
| Undergraduate Degree | 13 (36.1%) |
| Graduate Degree | 14 (38.9%) |
| Professional Degree (e.g., nurse, pharmacist etc.) | 9 (25.0%) |
| **Response Rate (based on 45 recruited participants)** | |
| Round 1 | 36 (80.0%) |
| Round 2 | 33 (73.3.%) |
| Round 3 | 30 (66.6.%) |

[*]Totals may exceed 100 as participants could select more than one status.

† The number of respondents who completed at least one Delphi round. Three additional people agreed to participate in the study but did not participate in either Delphi round.

‡ The number of respondents who completed at least one Delphi round. Nine additional people agreed to participate in the study but did not participate in any Delphi round.

SD = Standard Deviation.

Healthcare stakeholder respondents were also predominantly (83%) female, and all of these participants had at least some post-secondary education (100% bachelor's degree or higher) (Table 2). At the time of the study, these participants had worked, on average, 15.8 years in the continuing care sector. Half of these participants worked as decision makers in regional programs. The remaining participants were employed by provincial governments or worked in continuing care as planners or care providers.

## Delphi findings

The final Delphi results are shown in Table 3. Care partners and healthcare stakeholders reached consensus (high or very high agreement) on ten and three factors, respectively. All three factors identified by the healthcare stakeholders were also selected by the care partners. These three factors were all at the mezzo-level of the Health-Related Safety Framework [54], and were organized into social networks (having strong relationships and links with family, friends, and the community), the structure of the home and neighborhood (living in a safe home environment, e.g., with enough safety aids and equipment), and medical services (having coordinated care between all types of formal health care providers, e.g., physicians, home care workers, social workers) (Fig 1).

Care partners reached consensus on an additional 7 factors at the micro- (e.g., having enough money to afford to stay successfully in the community), mezzo- (e.g., having programs that help people to cope with mental health challenges such as anxiety, depression and loneliness), and macro-levels (e.g., ensuring that community-based alternatives to nursing home use are affordable) (see Table 3 and Fig 1). The complete results of each Delphi round are provided in S1 and S2 Tables, for care partners and healthcare stakeholders, respectively.

## Discussion

This study compares and contrasts how care partners and healthcare stakeholders prioritize the community-based supports, services, and personal strategies required to help older adults successfully age in place. Out of 27 potential factors, each participant group independently identified the same three factors (keeping one's home safe, maintaining strong inter-personal relationships, and coordinating care across formal providers) that they felt were very important to help older adults remain successfully in their community. While healthcare stakeholders did not reach consensus on other factors, care partners agreed that seven additional factors are also very important to consider. These findings help to establish community-based AIP priority areas and future research directions, guided in part by different participant perspectives.

### Priority areas identified by both participant groups

Care partners and healthcare stakeholders in this study agreed that having a safe home environment, maintaining strong social networks with others, and having coordinated healthcare services are very important to help older adults successfully age in place. While researchers have used qualitative approaches [43, 45, 62] and statistical models [44, 63, 64] to identify pertinent AIP factors, to the best of our knowledge this study is the first to identify highly important community-based AIP factors prioritized across participant groups. Our results align with the findings from a UK-based scoping review that defines the kinds of care and support (e.g., social relationships and activities, help related to mobility) that older adults need [65]. These identified priorities are also in keeping with others who have reported that successful AIP relies on adequate home safety (e.g., ensuring that people have the appropriate grab rails, non-slip surfaces and ramps in place) [66], and that people's decisions to remain in their community are based in part on their ability to effectively adapt their home [67]. Maintaining strong social networks is thought to be essential for successful AIP [68], and alternatively loneliness has been identified as an independent predictor of nursing home admission [69, 70]. Smaller scale interventions addressing these factors have been implemented, including environmental audits to help to improve home safety [71], activity and discussion groups to enhance social connection [72], and navigators to help older adults with chronic conditions transition through the healthcare system [73]. Findings from these smaller-scale interventions can help

**Table 3. Final consensus results by factor and participant group.**

| Delphi Factor | Care Partners | | | Healthcare Stakeholders | | |
|---|---|---|---|---|---|---|
| | % of scores ≥ 8 | IQR[†] | Consensus Level | % of scores ≥ 8 | IQR[†] | Consensus Level |
| **Micro—Biological** | | | | | | |
| Keeping physically active | 70.8% | 3 | Moderate | 76.7% | 1 | Moderate |
| Being continent (with or without the use of continence aids) | 75.0% | 0.3 | Moderate | 16.7% | 1 | Low |
| **Micro–Psychological** | | | | | | |
| Not having significant behavioural or mental health disorders | 79.2% | 1.5 | Moderate | 73.3% | 1 | Moderate |
| Thinking of oneself as healthy | 54.2% | 1 | Low | 6.7% | 1 | Low |
| Maintaining a positive attitude, having a high self-esteem and/or sense of personal identity | 75.0% | 1.8 | Moderate | 23.3% | 1 | Low |
| Keeping mentally active | 92.0% | 2 | High | 46.7% | 1 | Low |
| **Micro—Other** | | | | | | |
| Being someone who prepares and plans for the future (e.g., participates in health promotion activities, plans financially for the future, develops new skills) | 75.0% | 0.3 | Moderate | 20.0% | 1 | Low |
| Having enough money to afford to stay successfully in the community | 84.0% | 2 | High | 70.0% | 1 | Moderate |
| **Mezzo–Social Network** | | | | | | |
| Having strong relationships and links with family, friends, and the community | **88.0%** | **2** | **High** | **90.9%** | **1** | **High** |
| **Mezzo–Home and Neighbourhood Structure** | | | | | | |
| Living in a safe home environment (e.g., with enough safety aids and equipment) | **84.0%** | **2** | **High** | **86.7%** | **0** | **Very High** |
| Having a home layout that is appropriate (e.g., the absence of stairs) | 79.2% | 1 | Moderate | 53.3% | 1 | Low |
| **Mezzo–Social Services** | | | | | | |
| Having accessible and affordable community-based services (e.g., adult education, recreation and support programs) | 70.8% | 2 | Moderate | 13.3% | 1 | Low |
| Having a resource (e.g., information call centre) that helps people make informed choices about health care services that are available to them | 45.8% | 1.3 | Low | 20.0% | 1 | Low |
| Having public transportation that is affordable, reliable and accessible | 66.7% | 3 | Moderate | 20.0% | 0 | Low |
| Having formal healthcare providers (e.g., physicians, home care workers) who are aware of community-based services | 83.3% | 1.5 | High | 76.7% | 0 | Moderate |
| Having training & education programs for informal caregivers | 58.3% | 1.5 | Low | 13.3% | 0 | Low |
| Having programs that help people to cope with mental health challenges (e.g., anxiety, depression, loneliness)* | 91.7% | 2 | High | 56.7% | 1 | Low |
| Having programs that provide support to complete household chores (e.g., shoveling, mowing grass, completing minor household repairs) and other daily tasks (e.g., banking, grocery shopping)* | 75.0% | 1.3 | Moderate | 60.0% | 1 | Moderate |
| **Mezzo–Medical Services** | | | | | | |
| Having coordinated care between all types of formal health care providers (e.g., physicians, home care workers, social workers). | **95.8%** | **0.5** | **High** | 80.6% | 2 | **High** |
| Having physicians who provide house-calls & home visits | 58.3% | 1 | Low | 10.0% | 1 | Low |
| Having medical professionals (e.g., nurse practitioners, pharmacists) who regularly check the # and type of medications people are taking | 79.2% | 2 | Moderate | 23.3% | 0 | Low |
| Having good communication between informal & formal caregivers | 75.0% | 1.3 | Moderate | 76.7% | 0 | Moderate |
| Ensuring that people have adequate access to important allied and medical services (e.g., glasses, dental care, affordable medications, physiotherapy)* | 79.2% | 1.5 | Moderate | 56.7% | 1 | Low |
| **Macro–Policy and Societal** | | | | | | |
| Having access to affordable housing | 91.7% | 2 | High | 63.3% | 1 | Moderate |
| Having policies that allow people to reside in the community with an acceptable level of risk | 37.5% | 2.3 | Low | 66.7% | 1 | Moderate |
| Having access to funds that help people purchase assistive technology (e.g., motorized wheelchairs) and/or to modify their home (e.g., put in a wheelchair ramp). | 100.0% | 1.5 | High | 13.3% | 0 | Low |

(*Continued*)

**Table 3.** (Continued)

| Delphi Factor | Care Partners | | | Healthcare Stakeholders | | |
|---|---|---|---|---|---|---|
| | % of scores ≥ 8 | IQR[†] | Consensus Level | % of scores ≥ 8 | IQR[†] | Consensus Level |
| Ensuring that community-based alternatives to nursing home use (e.g., supportive housing in Manitoba, lodge and supportive living in Alberta) are affordable* | 83.3% | 2 | High | 76.7% | 0 | Moderate |

[†] IQR–Interquartile Range is the difference between the 75th and 25th quartile of rating.

* Additional Factor added based on participant feedback after Delphi Round 1.

Note: Bolded results identify factors that both participant groups reached consensus on.

to guide the implementation and evaluation of larger scale AIP innovations including their sustainability and scale.

## Differences in expert group perspectives

In addition to the aforementioned three priority areas, care partners selected seven additional factors that they felt were very important to consider. These additional factors are dispersed across the micro- (keeping mentally active, having enough money to stay in the community), mezzo- (having mental health programs, ensuring providers are aware of community-based care options), and macro- (having access to affordable housing, having funds to purchase assistive technology, having appropriate community-based alternatives to nursing homes) levels of the Health-Related Safety Framework [54]. These additional factors highlight the importance of having the finances to afford the supports and services needed to stay in the community, of

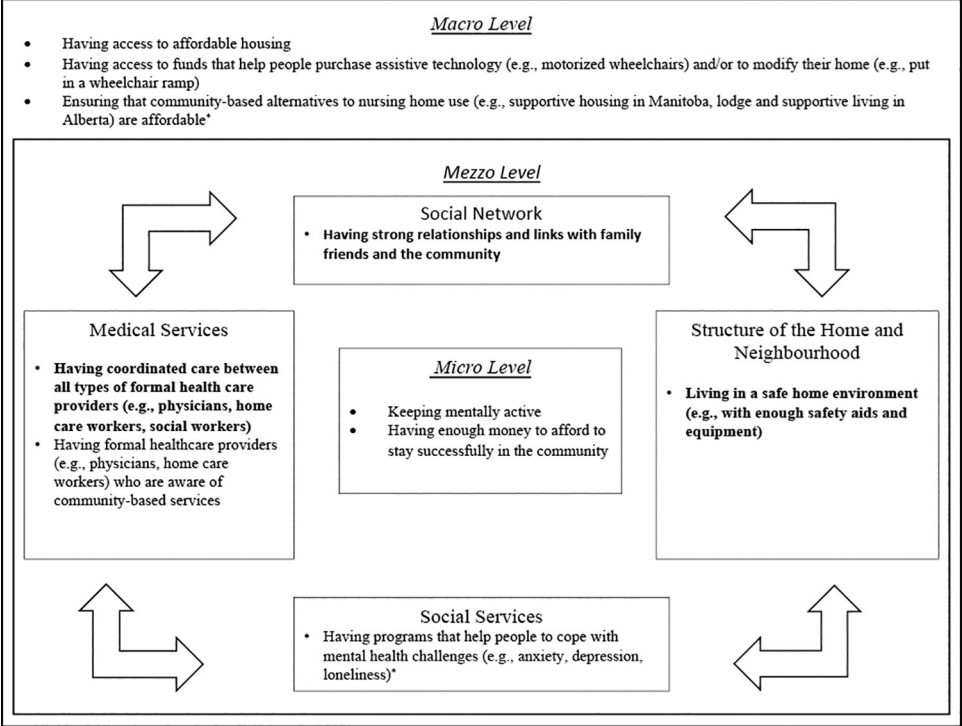

**Fig 1. Consensus factors categorized using the Health-Related Safety Framework [54].** Bolded text: Consensus was reached by both groups; Un-bolded text: Consensus was reached by care partners only. *Factor was added after Delphi Round 1.

ensuring that everyone has access to and is aware of these community-based programs, and the need for more mental health services. Overall, from the perspective and lived experience of care partners, successful AIP likely requires a larger number of diverse strategies to meet the varied needs of community-living older adults.

Other researchers have reported that different participant groups often have different perspectives about healthcare reform [25, 26, 36, 74, 75]. When investigating strategies to improve cancer care services in Greece, Efstathiou, Coll, Ameen and Daly (2011) reported that providers prioritized ways to better coordinate patient services, while healthcare users focused on treatments that would increase chances of survival [36]. Similarly, when investigating mental health research priorities, Owens, Ley and Aitken (2008) reported that clinicians and managers tended to prioritize research that focused on the provision of physical healthcare, while care partners and service users emphasized research that would promote independence, self-esteem and recovery [75].

Examining various perspectives and the potential reasons for their similarities and differences has value, given research showing that collaborative planning across stakeholder groups encourages different ways of thinking, greater reflexivity, and creates innovations that tend to have greater impact when implemented [27, 28, 76]. In the present study, healthcare stakeholders may have selected factors based on their scope of professional experience (e.g., selecting mezzo-level factors like coordinating formal healthcare, and arranging for safety aids and equipment). Alternatively, healthcare stakeholders in our study worked across the care continuum as policy makers, regional planners and providers, and these diverse perspectives may have limited their consensus making capacity. Conversely, care partners may have had more direct lived experiences and hence a broader understanding of the multidimensional nature of AIP, which influenced their selection of factors at the micro, mezzo, and macro-levels of the Health-Related Safety Framework [54]. Notwithstanding these potential explanations, the present study emphasizes the importance of engaging with 'end-users' when developing AIP research and care reform priorities, as one means to more effectively advance this agenda.

## Future research directions

Two sequential research directions are proposed. *First*, priorities identified in the present research are based on the views of experts from two Canadian healthcare regions. While our study findings have AIP reform implications for these local regions, further research is needed to define the extent to which they can be generalized to other jurisdictions. While there is some evidence to support the generalized nature of our study results (e.g., UK-based research identifies similar supports needed to age in place) [65], additional and ongoing research in this area is required.

*Second*, research is needed to examine the extent to which these key supports and services are provided successfully and equitably to various groups of community-dwelling older adults. Since Canada is not alone in its desire to enhance AIP, this analysis should ideally occur using an international comparative lens, in particular engaging with Denmark and Norway that are known to have well-developed home and community-based care programs [5]. The lessons learned from these comparisons can help to guide the development, feasibility testing, and eventual larger-scale implementation of innovations designed to better help older adults successfully age in place.

## Limitations

Strengths of this research include our rigorous use of the Delphi method, and our ability to create comparative results across participant groups. These strengths are offset by three

potential limitations. *First*, as previously mentioned, our results are based on expert opinion from two Canadian regions, and generalizations to other national or international jurisdictions, may be limited. *Second*, while participants were asked to rank the importance of each factor individually, it is likely that multiple factors interact to affect one's ability to AIP. We also did not ask participants to identity which of the listed supports and services were already available; it is feasible that participants, purposefully or unconsciously, rated factors based on perceived importance and availability. *Third*, we did not conduct an exhaustive literature review to identify all potential AIP supports, which may have influenced our study results. We minimized this third limitation by asking participants to provide additional factors during the first survey round.

## Conclusions

Healthcare stakeholders and care partners agreed that having a safe home environment, maintaining strong social networks with others, and having coordinated healthcare services are very important to help older adults successfully age in place. Care partners also reported that a larger and more diverse range of community-based factors are required to meet the varied needs of community-living older adults. These findings can help to prioritize community-based AIP reform initiatives. Future research should replicate these findings in other jurisdictions, examine the extent to which priority supports and services are available and accessible, and develop sustainable solutions to enhance their effectiveness.

## Supporting information

**S1 Table. Care partners full results from all rounds.**
(XLSX)

**S2 Table. Healthcare stakeholder full results from all rounds.**
(XLSX)

**S1 File. Healthcare stakeholder SPOR Delphi survey round 1.**
(PDF)

## Acknowledgments

We would like to acknowledge the contributions made by the following healthcare planners (Kimberly Weihs, Hana Forbes, Gina Trinidad, Douglas Faulder, Carol Anderson, Carmen Grabusic and Corinne Schalm) and community advisors (Connie Newman, Carol Draper, Louise Hutton, Lynne Mansell, Ingrid Crowther and Grant Geldart) who constitute our research team. These individuals contributed significantly to the manuscript by helping to refine the survey factors and actual questionnaire, and were instrumental in helping to recruit study participants. We would also like to thank Manitoba's and Alberta's SPOR Primary and Integrated Healthcare Innovation Networks for their help in recruiting participants.

## Author Contributions

**Conceptualization:** Megan Campbell, Tara Stewart, Heather Campbell-Enns, Andrea Gruneir, Erin Scott, Adrian Wagg, Malcolm Doupe.

**Formal analysis:** Megan Campbell, Tara Stewart, Thekla Brunkert, Andrea Gruneir, Gayle Halas, Matthias Hoben, Malcolm Doupe.

**Funding acquisition:** Malcolm Doupe.

**Methodology:** Megan Campbell, Tara Stewart, Erin Scott, Adrian Wagg, Malcolm Doupe.

**Project administration:** Megan Campbell.

**Supervision:** Malcolm Doupe.

**Writing – original draft:** Megan Campbell, Malcolm Doupe.

**Writing – review & editing:** Megan Campbell, Tara Stewart, Thekla Brunkert, Heather Campbell-Enns, Andrea Gruneir, Gayle Halas, Matthias Hoben, Erin Scott, Adrian Wagg, Malcolm Doupe.

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
