## [Decision Letter · Decision Letter 0]

2 Aug 2021

PONE-D-21-17416

Prioritizing supports and services to help older adults age in place: A Delphi study comparing the perspectives of family/friend care partners and healthcare stakeholders

PLOS ONE

Dear Dr. Campbell,

Thank you for submitting your manuscript to PLOS ONE. After careful consideration, we feel that it has merit but does not fully meet PLOS ONE’s publication criteria as it currently stands. Therefore, we invite you to submit a revised version of the manuscript that addresses the points raised during the review process.

Thank you for submitting this paper. Please see editor and reviewer comments below. I would like the authors to consider how their manuscript can be more relevant to an international audience in their resubmission. 

We look forward to receiving your revised manuscript.

Kind regards,

Anna Ugalde, PhD

Academic Editor

PLOS ONE

Journal Requirements:

Additional Editor Comments:

Thank you for submitting this paper. This is a good study and the methodology is sound. Two reviewer comments are attached. Please respond to these, in particular the areas of confusion in the results section as highlighted by Reviewer 2.

As identified by Reviewer 2, I would also ask the authors to reflect upon the importance of the comparison between Albert and Manitoba in Table 4. Please consider the relevance to an international audience. There is one brief line on the difference between these regions (line 155 page 7) but readers cannot be expected to understand the healthcare and health service differences between these regions. I would suggest a better description of these regions be included and their differences, alternatively, could this be removed from the manuscript. I am not sure the results are critically important given the low numbers. Please consider the best way for this to be managed.

Please also consider expanding the sections in the methods to be clear about how you selected your participants. The success of Delphi studies are contingent on the experts involved, we know very little about this participant group.

Thank you for the submission to Plos One.

Reviewers' comments:

Reviewer's Responses to Questions

**Comments to the Author**

1. Is the manuscript technically sound, and do the data support the conclusions?

Reviewer #1: Yes

Reviewer #2: Yes

2. Has the statistical analysis been performed appropriately and rigorously? 

Reviewer #1: Yes

Reviewer #2: Yes

3. Have the authors made all data underlying the findings in their manuscript fully available?

Reviewer #1: Yes

Reviewer #2: Yes

4. Is the manuscript presented in an intelligible fashion and written in standard English?

Reviewer #1: Yes

Reviewer #2: Yes

5. Review Comments to the Author

Reviewer #1: Thank you for conducting this thorough piece of research. I enjoyed reading it and I can see the value in this research.

My feedback is around making the manuscript and its findings a bit more relevant and useful for other countries. You have focused on Canada, which totally makes sense. But I would suggest you try to show what other countries, especially ones with a similar health system, are doing around this topic. Similarly, within Canada, what other states are doing could be relevant too. I see this being useful in the introduction as well as the discussion, more in the latter section than the former.

Furthermore, a few segments such as limitations of the study (I know you have it but in its current state, it reads like a superficial section), future research directions and especially relevance for practice and policy are missing. These sections make the ideas and arguments raised in this manuscript much more relevant to wider audience.

Reviewer #2: Overall a very interesting project with great importance. I do think there are some things that you can do to strengthen your manuscript.

Comment 1

I found myself confused about a few things. It seems that you analysed the data in three ways:

1. ratings were calculated using all participant data as a whole group (S1_Table)

2. As individual panels, i.e. as care partners and healthcare stakeholders

2. Comparing the two panels to each other.

I think this needs to be spelled out more clearly in the methods section and then more clearly labelled when discussing the results.

Comment 2

In the Study participants section (line 246) you say that the participants provided some feedback in at least one Delphi survey. Please state how many new participants there were in Round 2 so we can get a sense of the consistency of participants across rounds.

Comment 3

Results (this is related to comment 1). I am unclear what the primary results are – the two groups as a whole, the items that got consensus across both groups, or the results of the two separate groups. As I see it you had two aims – setting priorities and comparing the two areas. (The comparison has some issues = see next comment.) The way that my team do Delphi’s (Jorm AF. Using the Delphi expert consensus method in mental health research. Vol. 49, Australian and New Zealand Journal of Psychiatry. 2015), you would use the items that got consensus across both groups. Please be clearer in reporting your results.

Comment 4

Comparing the two groups of participants is interesting however given the numbers (n=between 10 and 20) the results are not very robust. A panel size of 23 or more is recommended. See Jorm citation above and Akins RB, Tolson H and Cole BR (2005) Stability of response characteristics of a Delphi panel: Application of bootstrap data expansion. BMC Medical Research Methodology 5: 37

You need to note this limitation.

Comment 5

Can you please explain more clearly why you removed some items from the Round 2 survey (I assume it is b/c they received high agreement in Round 1, but you haven’t said this in the methods section.

Comment 6

Some of your results do not add up. This may be because of the confusion explained in comments 1 and 3. Lines 276 and 277 say that 3 factors received high consensus from both panels but S1_Table has four items that received high agreement in Round one and 6 that received high agreement in round 2. The info in lines 276 and 277 seem to match table 3.

Related – in line 282 you say that care partners reached consensus on 7 factors but I count 10 in Table 3. Then you don’t mention that Health care stakeholders reached consensus on 3 factors.

Comment 7

Table 3 – this is minor but could you maybe reorganise this table so that the items are grouped in the same way as they are presented in Table 1?

Comment 8

Why were aged people not included in this study? They would have provided a very meaningful perspective. Please spell out why you did not include them and also note this as a limitation.

Comment 9

I am unclear why there were only 2 survey rounds for one group and 3 for the other. It seems that the write in options would need to be given the opportunity to be rated twice like all the other factors. Can you explain more clearly why this is and also note it as a limitation. Also, both groups should be given the opportunity to do all three rounds. I think you need to explain this more clearly in the body of the manuscript. I had to spend quite a bit of time looking at all the tables to get to only a low level of confusion.

Comment 10

With your second limitation you could say that this limitation was minimised by the fact that you allowed write in factors in the Round 1 survey.

6. PLOS authors have the option to publish the peer review history of their article (what does this mean?). If published, this will include your full peer review and any attached files.

Reviewer #1: No

Reviewer #2: **Yes: **Kathy Bond

---

## [Author Response · Author response to Decision Letter 0]

14 Sep 2021

As suggested by the Editor and identified as a limitation by Reviewer 2, we have removed text in this manuscript comparing Delphi responses across regions. Co-authors discussed including/excluding this text several times during manuscript development, and we appreciate the valuable insight and clarity that your comments have provided. Please see the following for major edits resulting from this change: (1) All original text about this topic has been removed, including in the methods (lines 171-173), results (page 20, lines 336-354) and discussion sections (lines 435-444), (2) Table 4 has been removed entirety (lines 351-354), and (3) We have further emphasized, as a future research direction, the need to replicate our study findings in different healthcare jurisdictions (see lines 77 in abstract, and lines 421-426 in the section entitled Future Research Directions). 

RESPONSES TO EACH OF THE REVIEWER’S COMMENTS

Reviewer #1 (R1)

R1 Comment 1: My feedback is around making the manuscript and its findings a bit more relevant and useful for other countries. You have focused on Canada, which totally makes sense. But I would suggest you try to show what other countries, especially ones with a similar health system, are doing around this topic. Similarly, within Canada, what other states are doing could be relevant too. I see this being useful in the introduction as well as the discussion, more in the latter section than the former.

We have approached edits to this comment cautiously, especially given Reviewer 2 comments (e.g., to more clearly align the stated study objectives and major findings). Please see the following:

1. As suggested, we have added text (a) in the Introduction showing that AIP is an international reform initiative (see lines 94-97), (b) showing that our results align with findings in the UK (lines 371-373) (c) in a newly created section called ‘Future Research Directions’ that proposes the need for national and international comparative research (see lines 420-433), and (d) emphasized in the ‘Limitations’ section a caution about generalizing study findings to other jurisdictions (see lines 448-450). 

2. Respectfully, however, we feel that a detailed discussion about specific AIP strategies used in different regions is beyond the scope of this paper. Our revised study purpose (lines 119-123) emphasizes the need to compare and contrast group opinions about what is needed, which is distinct from opinions about what works. We are happy to revise this decision if reviewers and/or editors feel otherwise.

R1 Comment 2: Furthermore, a few segments such as limitations of the study (I know you have it but in its current state, it reads like a superficial section), future research directions and especially relevance for practice and policy are missing. These sections make the ideas and arguments raised in this manuscript much more relevant to wider audience.

Thank-you for providing this clarity. We have done the following in response to this section (1) revised the text pertaining to study limitations (see text commencing line 445), and (2) included a new section entitled ‘Future Research Directions’ (see lines 420 to 433) that provides a more detailed description about future policy-related research implications. 

Reviewer #2 R2

R2 Comment 1: I found myself confused about a few things. It seems that you analysed the data in three ways:

1. ratings were calculated using all participant data as a whole group (S1_Table) 

2. As individual panels, i.e. as care partners and healthcare stakeholders

3. Comparing the two panels to each other. 

I think this needs to be spelled out more clearly in the methods section and then more clearly labelled when discussing the results.

Thank-you for this insightful comment. This feedback has helped us to streamline and clarify key aspects of the manuscript via the following edits:

1) We clarified that our study purpose (lines 119-123) was to compare and contrast important community-based AIP factors across study groups;

2) We removed all text pertaining jurisdictional comparisons (e.g., see lines 171-173; 336-348; 435-444 in revised manuscript); Table 4 has been removed in entirety (lines 351-354 in the revised manuscript). 

3) We have more clearly stated throughout the manuscript (e.g., see lines 137-139; 245; 254; 268-269) that results were collected and analyzed separately by study group (i.e., care partners and healthcare stakeholders), and;

4) We have amended a key typo with reference to our supplemental tables (see lines 230 and 324-325; S1 provides the final version of the survey, while S2 and S3 provides the full results, for the care partner and healthcare stakeholder study group, respectively). We apologize for any confusion that this has caused.

R2 Comment 2: In the Study Participants section (Line 246) you say that the participants provided some feedback in at least one Delphi survey. Please state how many new participants there were in Round 2 so we can get a sense of the consistency of participants across rounds. 

Line 246 is now line 279. We have clarified in revised lines 242-243; 279-280 and the revised footnotes provided in Table 2 (lines 303-306) that most Round 1 Delphi respondents participated in subsequent rounds, and that no new participants were added to the study after the first Delphi round. Those who dropped out, were no longer eligible to participate in subsequent rounds. 

R2 Comment 3: Results (this is related to comment 1). I am unclear what the primary results are – the two groups as a whole, the items that got consensus across both groups, or the results of the two separate groups. As I see it you had two aims – setting priorities and comparing the two areas. (The comparison has some issues = see next comment.) The way that my team do Delphi’s (Jorm AF. Using the Delphi expert consensus method in mental health research. Vol. 49, Australian and New Zealand Journal of Psychiatry. 2015), you would use the items that got consensus across both groups. Please be clearer in reporting your results.

Thank-you for the suggested reference. The major goal of the present study was to compare and contrast priority areas as identified by different expert groups. Please see the following edits designed to clarify our goal, our methods, and presentation of our results:

1) As discussed in response to your first comment, we have clarified and amended our purpose statement (see lines 51-54; 119-123), and edited select aspects of our Methods section (e.g., see lines 137-139; 245; 254; 268-269) to clarify how results were collected and analyzed separately for each group.

2) See lines 309-311 in the results section, which more clearly presents group-specific results and how these differed between expert groups. 

3) Our Discussion section has been framed to first discuss priority areas recommended by both participant groups (lines 366-384), and to then discuss differences in group-specific findings (lines 386-397). In two areas of the revised document (see lines 424-426; 464-467), we recommend that community-based AIP reform focus on the areas identified by both study groups, plus the additional factors identified by care partners. 

R2 Comment 4: Comparing the two groups of participants is interesting however given the numbers (n=between 10 and 20) the results are not very robust. A panel size of 23 or more is recommended. See Jorm citation above and Akins RB, Tolson H and Cole BR (2005) Stability of response characteristics of a Delphi panel: Application of bootstrap data expansion. BMC Medical Research Methodology 5: 37 

You need to note this limitation.

Please see our detailed response to this comment under the heading GENERAL EDIT 1. As recommended by the Editor, we have removed all reference to regional comparisons (e.g., in the purpose statement, methods, and results). The comparison is now focused only on family/friend care partners and healthcare stakeholders. 

R2 Comment 5: Can you please explain more clearly why you removed some items from the Round 2 survey (I assume it is b/c they received high agreement in Round 1, but you haven’t said this in the methods section.

We have added an explanation to the methods section, see lines 262-263.

R2 Comment 6: Some of your results do not add up. This may be because of the confusion explained in comments 1 and 3. Lines 276 and 277 say that 3 factors received high consensus from both panels but S1_Table has four items that received high agreement in Round one and 6 that received high agreement in round 2. The info in lines 276 and 277 seem to match table 3. 

Original lines 276-277 are now lines 309-311 in the revised manuscript and they have been edited to improve clarity. The results from these lines correspond to the data in Table 3. 

The line that mentions the complete results in the supplementary tables has been moved to the end of the paragraph (lines 324-325) to eliminate confusion as specific data in the supplementary files are not discussed in the paper. In addition, these files have been renamed. 

Related – in line 282 you say that care partners reached consensus on 7 factors but I count 10 in Table 3. Then you don’t mention that Health care stakeholders reached consensus on 3 factors. 

Line 282 is now line 320 in the revised manuscript. We have clarified in the paragraph before that the care partners reached consensus on 10 factors and healthcare stakeholder reached consensus on 3 factors (see line 309). 

R2 Comment 7: Table 3 – this is minor but could you maybe reorganise this table so that the items are grouped in the same way as they are presented in Table 1?

Thank you for this suggestion. The table has been reorganized so that the framework headings from Table 1 are also present in Table 3. In addition, all the factors have been organized in the same order as originally shown in Table 1 (see Table 3, pages 17-19). 

R2 Comment 8: Why were aged people not included in this study? They would have provided a very meaningful perspective. Please spell out why you did not include them and also note this as a limitation.

We have added a row in Table 2 (page 15) to indicate that 68% of the care partner participants were over age 65. Since the majority of our care partner participants were older adults, with many having been either formal or informal care recipients themselves, we feel that the older adult perspective is included in our study. We respectfully disagree however, that age itself is sufficient criteria to be considered an “expert,” therefore, we have not included it as a limitation. 

R2 Comment 9: I am unclear why there were only 2 survey rounds for one group and 3 for the other. It seems that the write in options would need to be given the opportunity to be rated twice like all the other factors. Can you explain more clearly why this is and also note it as a limitation. Also, both groups should be given the opportunity to do all three rounds. I think you need to explain this more clearly in the body of the manuscript. I had to spend quite a bit of time looking at all the tables to get to only a low level of confusion.

We have added text to the methods section to address why care partners completed two survey rounds (see lines 263-268). Since the goal of our study was to identify only the most important factors, we decided the care partner group had reached this goal after two rounds. 

R2 Comment 10: With your second limitation you could say that this limitation was minimised by the fact that you allowed write in factors in the Round 1 survey.

Thank you for the suggestion, see lines 45

---

## [Editor Report · Decision Letter 1]

19 Oct 2021

Prioritizing supports and services to help older adults age in place: A Delphi study comparing the perspectives of family/friend care partners and healthcare stakeholders

PONE-D-21-17416R1

Dear Dr. Campbell,

We’re pleased to inform you that your manuscript has been judged scientifically suitable for publication and will be formally accepted for publication once it meets all outstanding technical requirements.

Kind regards,

Anna Ugalde, PhD

Academic Editor

PLOS ONE
---

## [Editor Report · Acceptance letter]

29 Oct 2021

PONE-D-21-17416R1 

Prioritizing supports and services to help older adults age in place: A Delphi study comparing the perspectives of family/friend care partners and healthcare stakeholders 

Dear Dr. Campbell:

I'm pleased to inform you that your manuscript has been deemed suitable for publication in PLOS ONE. Congratulations! Your manuscript is now with our production department. 

Kind regards, 

on behalf of

Dr. Anna Ugalde 

Academic Editor

PLOS ONE